# Toward a Theology of the Imagination with S.T. Coleridge, C.S. Lewis, and J.R.R. Tolkien

**David Russell Mosley**

Independent Scholar, Concord, NH 03301, USA; mosley.david87@gmail.com

**Abstract:** While many authors continue to use terms like Christian Imagination or Sacramental Imagination, few seek to define what the term imagination means. In this paper, the author presents his findings based on a close reading of S.T. Coleridge, C.S. Lewis, and J.R.R. Tolkien. Rather than relying either on the definition of imagination as the ability to hold images in one's head, or the definition by which is meant creativity, this paper puts forward a synthesis of the positions of the three authors listed above. In the end, this paper concludes that the imagination is inherently connected to the divine act of Creation, which aids in clearing away the lenses of sin and familiarity.

**Keywords:** imagination; theology; poetry

## 1. Introduction

What is the imagination? While many people from poets to literary authors to philosophers and theologians have employed the term imagination, few seek to give it a proper definition. It is clear that few continue to use it in the ancient and medieval sense. In this sense, the imagination is the ability to hold an image in one's mind. Nor, however, is it clear that most use it to mean our ability to be creative as when we say of a child, "He has an overactive imagination", or "What a wonderful imagination you have". In these senses, we mean the child creates new ideas and images. But how, then, is the term employed, especially by those who wish to talk of a Romantic Imagination, a Christian Imagination, a Catholic Imagination, or even a Sacramental Imagination? It seems necessary, therefore, to endeavor a definition, and particularly a theological definition of imagination before we can begin asking if it can be peculiarly Christian or Catholic or anything else. What is more, once a definition of the imagination is essayed, and even in the process of reviewing the works of Coleridge, Lewis, and Tolkien, one needs to ask how does the imagination help form morality?

### 1.1. Coleridge

One of the early figures in redefining the word imagination, from the ability to hold images in our minds, to something else, is Samuel Taylor Coleridge (for a thorough treatment of Coleridge as a Christian source see (Guite 2017)). Coleridge's early flirtations with Unitarianism and his lifelong addiction to opium notwithstanding, he nevertheless is one of the founding figures of English Romanticism. Toward the end of his life, Coleridge wrote a fascinating autobiography. Rather than simply focusing on the events of his life, this *Biographia Literaria* focuses on the literary influences and accomplishments of his life. While Chapter 13 is the primary chapter that deals with the issue of the imagination, there is a passage earlier on that warrants examination.

Coleridge writes about the ability to understand the symbols inherent in the world around us. He gives as an example the air-sylph, a horned fly. He notes that the caterpillar as it forms its chrysalis, "leave room in its involucrum for antennae yet to come" (Coleridge 2008). His point is that these symbols already exist in the world. It is, to that end, our job to discover what these symbols are and what they mean. He continues:

In short, all the organs of sense are framed for a corresponding world of sense; and we have it. All the organs of spirit are framed for a correspondent world of spirit: though the latter organs are not developed in all alike. But they exist in all, and their first appearance discloses itself in the moral being. How else could it be, that even worldlings, not wholly debased, will contemplate the man of simple and disinterested goodness with contradictory feelings of pity and respect? 'Poor man! He is not made for *this* world.' Oh! Herein they utter a prophecy of universal fulfillment; for man must either rise or sink".

(Coleridge 2008)

But the question remains, who set these symbols? As Malcolm Guite rightly articulates the crux of this question: Is it humanity, with our own imagination? Is it God alone in a single act of creation long since past? Or are these symbols created by a continuous meeting between His imagination and ours?" (Guite 2008).

The answer, Guite argues, for Coleridge, is yes to the final question. These symbols arise out of a union between the infinite act of God and our finite acts. Guite sees this through the lens of God's creative act being an act of poetry (Guite 2008). He suggests that if this is so, then humanity's experience in writing poetry may have something "to teach us about God and the world" (Guite 2008).

Poetry allows the poet, and the reader, to both discover links between the world of symbols and meaning, as well as, in a lesser capacity, to create those links. It is similar to a saying attributed to Michelangelo, "The sculpture is already complete within the marble block, before I start my work. It is already there, I just have to chisel away the superfluous material". There is an interplay between the artist and the medium, discovering and uncovering, not simply imposing. But poetry is also in the work of awakening.

In the course of the *Biographia*, Coleridge discusses the origin of the work that began the English Romantic Movement, the *Lyrical Ballads*, written mostly by William Wordsworth, but with key contributions, not least of which being "The Rime of the Ancient Mariner" by Coleridge. Coleridge set himself the task of writing about things supernatural, "so as to transfer from our inward nature a human interest and a semblance of truth sufficient to procure for these shadows of imagination that willing suspension of disbelief for the moment, which constitutes poetic faith" (Coleridge 2008). Wordsworth, on the other hand, was to write of everyday issues, but "to give the charm of novelty" to them, and to excite a feeling analogous to the supernatural, by awakening the mind's attention from the lethargy of custom, and directing it to the loveliness and the wonders of the world before us; an inexhaustible treasure, but for which in consequence of the film of familiarity and selfish solicitude we have eyes, yet see not, ears that hear not, and hearts that neither feel nor understand (Coleridge 2008).

Poetry, whether of the everyday or supernatural, thus serves to awaken the mind, to clear the cobwebs of habit and custom. What makes this possible, however, is Coleridge's understanding of the imagination.

Coleridge divides the imagination in two. "The primary imagination", he writes, "I hold to be the living power and prime agent of all human perception, and as a repetition in the finite mind of the eternal act of creation in the infinite I AM" (Coleridge 2008). Our own primary imagination, therefore, is an organ of perception, it is precisely how we see the world around us. Malcolm Guite notes that the primary imagination exists not just in us, but, since it is a "repetition" of the "act of creation" it exists within the world itself. But what distinguishes our primary imagination and God's? Guite describes it this way:

In God the living power of Primary Imagination actually causes things to exist; indeed, it is the Logos 'through whom all things were made'. In the human being the Primary Imagination is the living power whereby all things, including humanity itself, are perceived. Because our Primary Imagination is a repetition in our finite mind of God's eternal act of creation, it enables us so to read God's work as to glimpse through them the mind of their Maker—-unless, of course, we perversely choose to refuse that glimpse, refuse to hear 'that

eternal language', which 'God utters', just as we might choose to describe our own language entirely in terms of its physicality and not in terms of its meaning".

([Guite 2008](#))

So, it is first through our organ of perception—-not that the imagination is a physical organ, but as Coleridge notes above it is a spiritual organ—-that we work with God to perceive, give, and discover meaning in the world around us. But what, then, of the secondary imagination?

For Coleridge, the secondary imagination, is "an echo of the former, coexisting with the conscious will, yet still as identical with the primary in the kind of its agency, and differing only in degree, and in the mode of its operation" ([Coleridge 2008](#)). The secondary imagination coincides with what we call the creative or poetic imagination. It this imagination we use when we write stories or poetry, paint pictures, etc. Coleridge does seem to limit it primarily to the fine and literary arts. A shortcoming, for sure. Nevertheless, this sense of the secondary imagination as an "echo" of the primary is, in many ways, ground-breaking. Coleridge is here saying that our acts of imagination participate in and are imitations of God's creative act. This is how poetry can awaken, clear the film of familiarity.

It is the last claim that is perhaps most interesting. If part of what poetry does is clear the film of familiarity, then it stands to reason this has, to some extent, always been its mission. It has become common to speak of the modern age as disenchanted. Perhaps one begins with Charles Taylor's notion of the buffered vs. porous selves, or perhaps one begins with the enlightenment, or perhaps one takes it even further back to someone like William of Ghent, William of Ockham, or John Duns Scotus. At present, I am not interested in genealogies of modernity. This, it seems to me, is secondary to the issue that the age in which we live has need of poetry to remove the film of familiarity, to aid us in seeing the world around us more clearly. Yet, this must always have been the case. Dante must have been concerned with this, so too the *Beowulf* poet, and Virgil and Homer too. There may be special issues we currently deal with that earlier ages did not, but there were issues in their own age that their poets helped them understand.

What is more, this understanding of the imagination shows the role it plays in forming morality. Only by understanding the imagination, as Coleridge sees it, as an imitation of and participation in God's creative act, can we understand that our works of imagination help form our understanding of the Good. For if we are, as Coleridge suggests, truly participating in God's creative act, then our own acts of creativity must, to some extent, also be in line with the Good.

Coleridge, however, is by no means the last to begin this re-working of imagination, nor its implications for what imaginative works may do.

### 1.2. C.S. Lewis

C.S. Lewis in many ways can be said to be following in Coleridge's footsteps. Not only as one who continued to contribute to a kind of expanded Christian Romanticism, one that did not simply look to nature, but beyond it as well, but as one continuing the unique understanding of imagination that Coleridge was working with. It does seem, however, that Lewis had not the same respect for the imagination as Coleridge did. Before moving on, I want to look at an early poem written by Lewis.

Much of Lewis's unpublished poetry was collected in one edited volume by Walter Hooper, who served as Lewis' secretary before his death. Many of the poems were unnamed and so Hooper took it upon himself to name them, and occasionally, to give rough estimates as to their dates. This only matters insofar as it appears that Lewis's poem, "Reason" (so named by Hooper) was likely written before Lewis's conversion.

The poem explores the relationship between reason and imagination. Reason is depicted as bright, virginal, celestial. Imagination, on the other hand, is dark yet beautiful. She is warm too, but obscure, and infinite, the daughter of night. There, Lewis clearly sees imagination as related to dreams. But here also, we see that imagination is not simply the holding of images in one's head, nor it is simply creative

invention. It is something else, something twinned, but opposite, to reason. In the latter half of the poem, Lewis writes:

> Tempt not Athene. Wound not in her fertile pains
> Demeter, nor rebel against her mother-right.
> Oh who will reconcile in me both maid and mother,
> Who make in me a concord of the depth and height?
> Who make imagination's dim exploring touch
> Ever report the same as intellectual sight?
> Then could I truly say, and not deceive,
> Then whole say, that I BELIEVE (Lewis [1964] 1992).

Lewis is showing us that reason is related to imagination. They seem to perform similar functions, but in different modes, or on different wavelengths. The key too seems not to give one or the other the head, but to get them working in concord.

Lewis's own conversion story plays this out. In *Surprised by Joy*, Lewis notes that the things he most loved, prior to his conversion, the myths and stories of the North, he believed to be nothing more than beautiful lies, breathed through silver as he would describe it to J.R.R. Tolkien and Hugo Dyson one night on Addison's Walk. Yet the things he believed most true, that the world exists according to materialistic and atheistic principles, filled him with dread. After his conversion, and thanks in no small part to Tolkien's poem "Mythopoeia" which will be examined below, Lewis sees a higher place for the imagination, but it is not clear he sees it on the same level as Coleridge. Part of this may have to do with Lewis's early flirtations with the occult, but I do not want to get bogged down in psychoanalyzing the dead.

In his essay, "Bluspels and Flalansferes", Lewis discusses the issues of metaphor. His concern is to show that metaphors are at once built-in to the very words we use on a daily basis, but that new ones may also be created. He writes:

> To speak more plainly, he who would increase the meaning and decrease the meaningless verbiage in his own speech and writing, must do two things. He must become conscious of the fossilized metaphors in his words; and he must freely use new metaphors, which he creates for himself. The first depends upon knowledge and therefore on leisure; the second on a certain degree of imagination.
>
> (Lewis 2018)

Here we can see, to some extent, the influence of Coleridge, though it is also likely the influence of Owen Barfield as well. Namely, just as Coleridge understood the imagination to be an organ of perception, one that works with God to perceive the meaning of things, here Lewis suggest that one who wishes to speak or write well must both work with words as they are and create new meanings. While leisure is necessary for the proper consideration of the metaphors inherent in our words, imagination is necessary to create new metaphors.

Lewis argues that imagination, rather than being the organ of perception, as Coleridge does, or the organ of truth, which he says is reason, is the organ of meaning. He even goes on to say that it is the poets who rank highest in this regard, and that the imagination itself has a kind of truth inherent to it:

> It will have escaped no one that in such a scale of writers the poets will take the highest place; and among the poets those who have at once the tenderest care for old words and the surest instinct for the creation of new metaphors. But it must not be supposed that I am in any sense putting forward the imagination as the organ of truth. We are not talking of truth, but of meaning: meaning which is the antecedent condition both of truth and falsehood, whose antithesis is not error but nonsense. I am a rationalist. For me, reason is the natural organ of truth; but imagination is the organ of meaning. Imagination, producing new metaphors or

revivifying old, is not the cause of truth, but its condition. It is, I confess, undeniable that such a view indirectly implies a kind of truth or rightness in the imagination itself.

(Lewis 2018)

Lewis's distinction seems to suggest a relationship between his own understanding of imagination and Coleridge's. While Lewis does not call the imagination an organ of perception he does say it is an organ of meaning. Meaning, for Lewis, is a mercurial trait.

In his book, *That Hideous Strength*, the final book in his cosmic trilogy, there is a scene where the oyarses, baptized versions of the Greco-Roman gods who pilot the planets through the heavens, descend upon St. Anne's Manor. They descend directly on Ransom and Merlin, and indirectly on those who waiting, one might say hiding, in the kitchen. Whereas in the kitchen, the company is spending their time in making plays on words, or with words, and putting forth ideas jokingly that seem quite serious on reflection, things go differently in the Blue Room where Ransom and Merlin wait (Lewis 2003).

Upstairs this first change had a different operation. There came an instant at which both men braced themselves. Ransom gripped the side of his sofa; Merlin grasped his own knees and set his teeth. A rod of coloured light, whose colour no man can name or picture, darted between them: no more to see than that, but seeing was the least part of their experience. Quick agitation seized them: a kind of boiling and bubbling in mind and heart which shook their bodies also. It went to a rhythm of such fierce speed that they feared their sanity must be shaken into a thousand fragments. And then it seemed that this had actually happened. But it did not matter: for all the fragments—-needle-pointed desires, brisk merriments, lynx-eyed thoughts—-went rolling to and fro like glittering drops and reunited themselves. It was well that both men had some knowledge of poetry. The doubling, splitting, and recombining of thoughts which now went on in them would have been unendurable for one whom that art had not already instructed in the counterpoint of the mind, the mastery of doubled and trebled vision. For Ransom, whose study had been for many years in the realm of words, it was heavenly pleasure. He found himself sitting within the very heart of language, in the white-hot furnace of essential speech. All fact was broken, splashed into cataracts, caught, turned inside out, kneaded, slain, and reborn as meaning. For the lord of Meaning himself, the herald, the messenger, the slayer of Argus, was with them: the angel that spins nearest the sun. Viritrilbia, whom men call Mercury and Thoth.

(Lewis 2003)

Mercury in the medieval and ancient view is the lord of Meaning, and Lewis uses that here and elsewhere in his writings. In the kitchen, metaphor seems to reign, but here it is not metaphor, but meaning itself, the very thing metaphors, the very thing imagination, is meant to aid us in receiving is felt direct by those in Mercury's presence. In fact, the kitchen gives us the role of the imagination as the organ of meaning. The imagination, in Lewis's conception is the mediation of meaning through something else. And this is not just true for the coming of Mercury, but is true for all the mediated presences of the planets.

As noted above, Lewis does not seem to have valued the imagination in quite the same way that Coleridge did, nor, as I will show next, as Tolkien did. He seems to have been more concerned with its ability to mislead. However, it is also clear that he saw its usefulness in explaining not simply theological truths, but moral truths as well. The Good can be found through the acts of bravery and kindness that so often appear in Lewis's fiction. Nevertheless, this notion of the imagination as the organ of meaning, as opposed to truth, shows Lewis understood it to be necessary. Ransom and Merlin are almost torn apart by the coming of Mercury, and it is their knowledge of poetry, an art usually attributed to Mercury's older brother and god of the sun, Apollo. It is the familiarity with metaphor and imagination that allows them, in part, to survive. When writing a review of Tolkien's *Lord of the Rings*, Lewis says "The value of myth is that it takes all the things we know and restores to them the

rich significance which has been hidden by 'the veil of familiarity'" (Lewis 1982). In that last line, we should recognize a reference to Coleridge. But what Coleridge reserved for poetry Lewis expands to myth. Most myths are themselves written as poetry, but the work Lewis is discussing, while filled with poetry, is a large work of prose.

### 1.3. J.R.R. Tolkien

After their trip down Addison's Walk, Tolkien went home and wrote a poem to Lewis. He called the poem, "Mythopoeia" and addressed it to Misomythus, calling himself Philomythus. Lewis was the myth-hater and Tolkien its lover because Lewis did not then understand what myths were. In the poem, Tolkien highlights the importance of story, myth, or imagination in the shaping and perceiving of the world around us. He writes:

> Yet trees and not 'trees', until so named and seen -
> and never were so named, till those had been
> who speech's involuted breath unfurled,
> faint echo and dim picture of the world,
> but neither record nor a photograph,
> being divination, judgement, and a laugh,
> response of those that felt astir within
> by deep monition movements that were kin
> to life and death of trees, of beasts, of stars:
> free captives undermining shadowy bars,
> digging the foreknown from experience
> and panning the vein of spirit out of sense.
> Great powers they slowly brought out of themselves,
> and looking backward they beheld the Elves
> that wrought on cunning forges in the mind,
> and light and dark on secret looms entwined. (Tolkien [1964] 2001)

They are humans, people who named and thus gave and discovered the shape and meaning of things. Both Lewis and Tolkien were influenced on this front by Owen Barfield, but there is not space to discuss him further at present. The key here is how Tolkien helped show Lewis what myth, and ultimately what imagination, is capable of. Tolkien shows us that myth and imagination allow us to actually see the world around us. Later in the poem he writes:

> He sees no stars who does not see them first
> of living silver made that sudden burst
> to flame like flowers beneath the ancient song,
> whose very echo after-music long
> has since pursued. There is no firmament,
> only a void, unless a jewelled tent
> myth-woven and elf-patterned; and no earth
> unless the mother's womb whence all have birth. (Tolkien [1964] 2001)

Tolkien argues again and again that sight and true sight, is a necessary outcome of the imagination. He also argues in the poem that this right of what he will call fantasy, is one inherent to us, even when we misuse it. For Tolkien understands that myths can become corrupted. We can create gods and then bow down to worship them. Imagination may have truth inherent in it, and may be the organ of both meaning and perception, but it can also lead us astray. But so too can reason. Cold logic, unmoved by beauty can lead to atrocities far worse than the imagination can. And this is central to Tolkien's understanding of the moral dimension of the imagination. An act of sub-creativity done for the wrong reasons can still instruct and delight, but only because it cannot completely remove itself from its

ultimate source in God. We have the right to make myths, but we must beware that we do not turn our statues into gods and then bow down to them.

In his essay, *On Fairy-stories*, Tolkien links this right to create to the reality that we our created in the image and likeness of the Creator. (Tolkien 2014). But unlike Lewis or Coleridge, Tolkien gives a particular name to this act of creativity beyond the imagination. Tolkien calls it sub-creation. The idea being that only God, properly speaking creates, for to create means to create from nothing. Humans, and other rational beings, cannot create, but we can take what God has made from nothing and make new things out of it. This is sub-creation. For Tolkien, fantasy, rather than poetry or realist fiction, is the more sub-creative (Tolkien 2014). Alison Milbank writes on this understanding in Tolkien:

> And it is in the ability to create—-fiction is linked to the Latin vert facere, to make—-that the artist comes closest to God. For us to recognize the world as God's creation, we have to see it as a work of art; for us to recognize the creative power of the artist, we similarly have both to experience his or her fiction as a world but also be aware of its constructed nature.
>
> (Milbank 2009)

This imitative and participatory relationship between the writer, and in fact one can expand this to all human creativity, and God is one that allows us to see creation, both to see it as an act of creation, and thus also as a gift. For Tolkien, this is called recovery.

Recovery, for Tolkien, is a recovery of vision. He writes:

> Recovery (which includes return and renewal of health) is a re-gaining—-regaining of a clear view. I do not say "seeing things as they are" and involve myself with the philosophers, though I might venture to say "seeing things as we are (or were) meant to see them"—-as things apart from ourselves. We need, in any case, to clean our windows; so that the things seen clearly may be freed from the drab blur of triteness of familiarity—-from possessiveness.
>
> (Tolkien 2014)

Tolkien will not, in this essay, give the declarative statement as Coleridge does, that imagination, here under the guise of fantasy, leads to a recovery of seeing things as they are, yet that is clearly where he is headed. The imagination, just as Coleridge and Lewis have both argued, leads to a better understanding of reality. Tolkien's sense of recovery, linked with his sense of sub-creation as an imitative and participatory act, link him directly back to Coleridge. Note too the final line of the second sentence. What the imagination does for us is allow us to see things outside of ourselves and things apart from ourselves. A mechanistic or utilitarian view of reality sees the world only in relation to how it can benefit me. The Imagination teaches us to see trees not simply for how we might make a table or a home our of them, but as beings in and of themselves, endowed with reality by the same God who made us. Yet how are we to understand the differences about the imagination among these three authors? And, what is more, how are we to move forward?

## 2. Synthesis

How then ought we to understand this word, so popular these days, imagination. An easy definition is perhaps not possible. But to borrow from the three authors surveyed we might be able to say that the imagination, in its key or primary function, is an organ of perception by which we discover and shape the meaning of the world around us. This perception, further, is a participation in the creative act of God. In its secondary function, then, the imagination is an organ of meaning by which we both come to an understanding of reality and through which we can do the sub-creative work of the "poetic imagination". Of course, none of this nullifies the older definition of the ability to hold images in one's head, but these additions augment the definition to show that this imaging is not purely passive, nor is it limited to holding or even re-creating and re-designing said images, but coming to an understanding of them.

This raises the question, however: is there something such as the Christian or Sacramental imagination? To fully answer this question would require yet another paper. But to begin an answer, I want to suggest that no, there is not a peculiarly Christian Imagination. The imagination, as I have defined it through Coleridge, Lewis, and Tolkien, is available to all. However, this does not mean that the Christian use of it is not peculiar. Christians do, or ought, to have a way of viewing reality unique to them, in that they are aware of what reality is. However, this does not preclude the possibility of non-Christians having access to this same imagination. Malcolm Guite shows this well in his investigation of writers like Thomas Hardy in *Faith, Hope and Poetry*.

Further, the imagination fully available to all, seems to be not only innate, but also to come on in ecstatic moments. Nearly all artists, but especially poets and literary authors, describe the process of creating as one of reception, as though something foreign to them comes upon them to draw out a work of art. Homer, Virgil, Dante, William Blake, Coleridge, Lewis, and Tolkien, all describe it in this way. One is reminded of the prophets of the Old Testament like Jeremiah who described the Word of Lord coming upon him like a fire in his stomach that he could not hold in if he tried. Ancient and Medieval mystics describe it this way as well. Julian of Norwich is on what she hopes is her deathbed when the visions come. So, while there may not be a peculiarly Christian or Sacramental imagination, it may well be that certain people are better attuned to reception. Many of those so attuned may be Christian, but many of those who are not will often be found to sing in harmony with the Christian understanding of reality.

Finally, the imagination helps us learn to see reality around us. This means seeing it as an act of gift, which means we must respond to it as such. Not trees, nor birds, nor stones, nor other people can be treated in a utilitarian fashion. Rather, we must come to recognize that this meaning-making in which we are involved means learning the Good. For the Good, the True, and the Beautiful are intertwined. Where you find one, you must, to some extent, find the others. Thus, as the imagination often leads us on the path of Beauty, on it we will find the True and the Good as well. And this is not to say that acts of the imagination must have moralizing impulses behind them. A story may have an intended moral, like Aesop's *Fables*, but the imagination and its effects will also have something to teach us about the True and the Good as well as the beautiful.

## 3. Conclusions

This essay, by necessity, cannot plumb the depths of the imagination fully. I have focused on only three writers, writers in English no less, who have discussed this subject. There is much more to be said, more to be researched, and more voices to which we must listen. But the key, I believe, remains the same. The imagination is both an organ of perception and an organ of meaning. By it, when properly attuned, we come not only to understand reality, to see more clearly, to have washed away the film of familiarity, but we also come to participate in and imitate God as Creator. This imitation can only be sub-creative, but it nevertheless gives rise to truly new things in reality, things that did not exist before, things that ought to aid us in seeing reality more clearly. It is thus, circular, or perhaps better, spirated, re-covering the same ground but in new ways, so that an act of the imagination renders vision clearer and that clearer vision leads to new acts of the imagination. Perhaps, with such an understanding, we can come to an understanding of God, of the Real, even higher than that of the past. Perhaps too we can come into a better relationship with one another and the world around us.

**Funding:** This research received not external funding.

**Conflicts of Interest:** The author declares no conflict of interest

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
