# Peer review of "Toward a Theology of the Imagination with S.T. Coleridge, C.S. Lewis, and J.R.R. Tolkien"

_religions, doi:10.3390/rel11050238_

Round 1

Reviewer 1 Report

There are no footnote references with page numbers. What makes it confusing is that on line 214 there is a footnote 15 but as far as I can see no previous footnotes. Further, there is no footnote or endnote 15. This is very frustrating as I was interested in knowing more about Lewis's use of Mercury to denote meaning.

There seem to be a few minor glitches. On lines 81 and 82 there is a sentence that ends with ' act of creation'. Thing is there seems to be a need for a clause that answers to the "since" in the sentence. The sentence ends abruptly.

On line 55, and this is a bit pedantic, there is a split infinitive: " as well as to, in a lesser capacity, create those links." I would place the first comma after "as". 

In line 144 I think the "in" in between "his" and "goddesses" is out of place.

The author might supply a reference or two concerning the influence of Owen Barfield on Lewis and Tolkien in lines 254-5. 

Author Response

Author notes in red.

There are no footnote references with page numbers. What makes it confusing is that on line 214 there is a footnote 15 but as far as I can see no previous footnotes. Further, there is no footnote or endnote 15. This is very frustrating as I was interested in knowing more about Lewis's use of Mercury to denote meaning. There are no footnotes with page numbers because that is not the citation style asked for. I can, of course, provide these if necessary.

There seem to be a few minor glitches. On lines 81 and 82 there is a sentence that ends with ' act of creation'. Thing is there seems to be a need for a clause that answers to the "since" in the sentence. The sentence ends abruptly. Fixed.

On line 55, and this is a bit pedantic, there is a split infinitive: " as well as to, in a lesser capacity, create those links." I would place the first comma after "as". Infinitive unsplit.

In line 144 I think the "in" in between "his" and "goddesses" is out of place. In should have been inner

The author might supply a reference or two concerning the influence of Owen Barfield on Lewis and Tolkien in lines 254-5. While I can add more, I would prefer to leave this out for now as a line or two would not suffice.

Reviewer 2 Report

General comment

Overall, this is an interesting paper and contains a good deal of creative thinking. There some points listed below that require attention. What is not clear is why ‘Catholic’ (especially given that several authors mentioned in the essay (Malcolm Guite, for example) are not Catholic. [I am making this comment as someone who is a committed Roman Catholic]. Moreover, Orthodox Christians are even more than Catholics immersed in the imaginative way of relating to God/divinity and sacredness. Somehow, focusing on the Catholic imagination and saying very little about it (except that there is no space for further study of it) doesn’t contribute much to the paper. Perhaps just sticking to the idea of the Christian imagination (rather than specifying a particular denomination) would make more sense. Otherwise, there is a danger of making several unhelpful assumptions. Without a supportive evidence and solid argument references to Catholic or Christian are not helpful. The fact that some Protestant Churches have fewer ‘physical’ appeals to imagination in their liturgy doesn’t mean that they lack imagination. It would be dangerous to imply this and it would be simply untrue esp. as several key poets who heavily rely on imagination (f.e. RS Thomas) are not Catholic. This aspect of the paper requires substantial rethinking, clarification and more solid argumentation, What is being proposed about the imagination and its connection to the divine (esp. creation) doesn’t need to be reduced to any specific religious and Christian denomination. It is simply a theistic view of the reality – this point is interesting and well made thought the paper and the analysis of the three poets is solid. I enjoyed reading the paper. Thank you.

Specific points:

  • The text requires editing; there are occasional problems with syntax and punctuation (for example, 35 ‘His point, is that…’ – there is no need for a comma after ‘point’; plus this sentence can be broken into two in order to make it easier to read)
  • 4-5 There is a problem with the beginning of the Abstract. The first two sentences should be one.
  • 15-16 The distinction between poets, authors, philosophers and theologians is rather odd. Can’t poets or philosophers be also authors? Or does the term ‘authors’ apply to authors of literature? If so it should say ‘literary authors’ or something similar.
  • 47 There is a problem with the question here – should ‘be’ be ‘by’?
  • 102-103 What does ‘clear the film of familiarity’ mean for Coleridge? Does ‘film of familiarity’ differ from Lewis’ ‘veil of familiarity’? NB. The phrase (as it is not the author’s of this paper) should be in single quotation marks
  • 107 Why ‘porous’ starts with a capital ‘P’?
  • 122 This doesn’t read well: ‘To this I will turn momentarily. Before I do, however,…’
  • 143-144 This sentence not only sounds rather informal, the second part is unclear: ‘As Malcolm Guite has occasionally joked in his presentations on this poem, it sounds as though Lewis is saying he needs to get his in goddesses in order’.
  • 300 ‘distinctions’ – which ones exactly? Is it about the different perspectives on the imagination?
  • 321 ‘there is not time or space’ ?
  • 327 ‘Julian’ – Julian of Norwich?
  • 336 Adopting the language of organs in discussing the imagination can be problematic (‘organ of perception and an organ of meaning’) – perhaps quality or disposition (of perception or/and meaning) would be better? Surely, the imagination is not a physical organ. Or, perhaps the term ‘organ’ needs to be clarified or clearly explained.

Author Response

Author comments in read

Overall, this is an interesting paper and contains a good deal of creative thinking. There some points listed below that require attention. What is not clear is why ‘Catholic’ (especially given that several authors mentioned in the essay (Malcolm Guite, for example) are not Catholic. [I am making this comment as someone who is a committed Roman Catholic]. Moreover, Orthodox Christians are even more than Catholics immersed in the imaginative way of relating to God/divinity and sacredness. Somehow, focusing on the Catholic imagination and saying very little about it (except that there is no space for further study of it) doesn’t contribute much to the paper. Perhaps just sticking to the idea of the Christian imagination (rather than specifying a particular denomination) would make more sense. Otherwise, there is a danger of making several unhelpful assumptions. Without a supportive evidence and solid argument references to Catholic or Christian are not helpful. The fact that some Protestant Churches have fewer ‘physical’ appeals to imagination in their liturgy doesn’t mean that they lack imagination. It would be dangerous to imply this and it would be simply untrue esp. as several key poets who heavily rely on imagination (f.e. RS Thomas) are not Catholic. This aspect of the paper requires substantial rethinking, clarification and more solid argumentation, What is being proposed about the imagination and its connection to the divine (esp. creation) doesn’t need to be reduced to any specific religious and Christian denomination. It is simply a theistic view of the reality – this point is interesting and well made thought the paper and the analysis of the three poets is solid. I enjoyed reading the paper. Thank you.

Much of this language was left over from presenting at a Catholic conference. To aid with simplicity, the explicit references to Catholic Imagination have been excised and replaced simply with Christian.

Specific points:

  • The text requires editing; there are occasional problems with syntax and punctuation (for example, 35 ‘His point, is that…’ – there is no need for a comma after ‘point’; plus this sentence can be broken into two in order to make it easier to read) Fixed
  • 4-5 There is a problem with the beginning of the Abstract. The first two sentences should be one. 
  •  Fixed
  • 15-16 The distinction between poets, authors, philosophers and theologians is rather odd. Can’t poets or philosophers be also authors? Or does the term ‘authors’ apply to authors of literature? If so it should say ‘literary authors’ or something similar.
  •  Fixed
  • 47 There is a problem with the question here – should ‘be’ be ‘by’?
  •  Fixed
  • 102-103 What does ‘clear the film of familiarity’ mean for Coleridge? Does ‘film of familiarity’ differ from Lewis’ ‘veil of familiarity’? NB. The phrase (as it is not the author’s of this paper) should be in single quotation marks
  • The issue of single quotation marks has not been changed as this is not listed as part of the style guide and the American commonplace is to use double quotation marks only. As regards "film of familiarity" this is addressed throughout the paper.
  • 107 Why ‘porous’ starts with a capital ‘P’? 
  •  Fixed
  • 122 This doesn’t read well: ‘To this I will turn momentarily. Before I do, however,…’ 
  •  Fixed
  • 143-144 This sentence not only sounds rather informal, the second part is unclear: ‘As Malcolm Guite has occasionally joked in his presentations on this poem, it sounds as though Lewis is saying he needs to get his in goddesses in order’. Removed
  • 300 ‘distinctions’ – which ones exactly? Is it about the different perspectives on the imagination? Differences between the authors on the imagination
  • 321 ‘there is not time or space’ ? Line removed
  • 327 ‘Julian’ – Julian of Norwich? Yes
  • 336 Adopting the language of organs in discussing the imagination can be problematic (‘organ of perception and an organ of meaning’) – perhaps quality or disposition (of perception or/and meaning) would be better? Surely, the imagination is not a physical organ. Or, perhaps the term ‘organ’ needs to be clarified or clearly explained. Colerdige introduces this language in an early section of the essay. I have retained but included in the first instance an explicit statement that it is a spiritual and not physical organ.